# Rapid Assay for Sick Children with Acute Lung infection Study (RASCALS): diagnostic cohort study protocol

John Alexander Clark ![ORCID],[1,2] Iain Robert Louis Kean,[1] Martin D Curran,[3] Fahad Khokhar,[4] Deborah White,[2] Esther Daubney,[2] Andrew Conway Morris,[2] Vilas Navapurkar,[2] Josefin Bartholdson Scott,[4] Mailis Maes,[4] Rachel Bousfield,[2] Theodore Gouliouris,[5] Shruti Agrawal,[1,2] David Inwald,[2] Zhenguang Zhang,[1] M Estée Török ![ORCID],[5] Stephen Baker,[4] Nazima Pathan[1,2]

¹Department of Paediatrics, University of Cambridge, Cambridge, UK
²Cambridge University Hospitals NHS Foundation Trust, Cambridge, UK
³Clinical Microbiology and Public Health Laboratory, Cambridge, UK
⁴Cambridge Institute of Therapeutic Immunology and Infectious Disease, University of Cambridge, Cambridge, UK
⁵Department of Medicine, University of Cambridge, Cambridge, UK

**Correspondence to**
Dr John Alexander Clark;
jac302@cam.ac.uk

## ABSTRACT

**Introduction** Lower respiratory tract infection (LRTI) is the most commonly treated infection in critically ill children. Pathogens are infrequently identified on routine respiratory culture, and this is a time-consuming process. A syndromic approach to rapid molecular testing that includes a wide range of bacterial and fungal targets has the potential to aid clinical decision making and reduce unnecessary broad spectrum antimicrobial prescribing. Here, we describe a single-centre prospective cohort study investigating the use of a 52-pathogen TaqMan array card (TAC) for LRTI in the paediatric intensive care unit (PICU).

**Methods and analysis** Critically ill children with suspected LRTI will be enrolled to this 100 patient single-centre prospective observational study in a PICU in the East of England. Samples will be obtained via routine non-bronchoscopic bronchoalveolar lavage which will be sent for standard microbiology culture in addition to TAC. A blood draw will be obtained via any existing vascular access device. The primary outcomes of the study will be (1) concordance of TAC result with routine culture and 16S rRNA gene sequencing and (2) time of diagnostic result from TAC versus routine culture. Secondary outcomes will include impact of the test on total antimicrobial prescriptions, a description of the inflammatory profile of the lung and blood in response to pneumonia and a description of the clinical experience of medical and nursing staff using TAC.

**Ethics and dissemination** This study has been approved by the Yorkshire and the Humber-Bradford Leeds Research Ethics Committee (REC reference 20/YH/0089). Informed consent will be obtained from all participants. Results will be published in peer-reviewed publications and international conferences.

**Trial registration number** NCT04233268.

## Strengths and limitations of this study

► This study describes clinical application of a customisable rapid diagnostic respiratory microarray in critically ill children.
► The sampling technique for this project is non-bronchoscopic bronchoalveolar lavage, which is a reliable sampling method for deep respiratory samples, while previous studies have utilised upper airway samples, or endotracheal tube secretions which may be prone to contamination.
► Following this single-centre study, a wider multi-centre evaluation will be required to determine feasibility and cost efficiency of this diagnostic method.
► There is no universally agreed diagnostic criteria for pneumonia in children, therefore, clinical suspicion of pneumonia by the treating team rather than a specific set of diagnostic criteria are being used for enrolment.

## INTRODUCTION

Lower respiratory tract infection (LRTI) is a leading cause of admissions to paediatric intensive care units (PICUs) in the UK, and greatest worldwide cause of mortality in young children.[1 2] PICU physicians regularly prescribe antimicrobial therapy for LRTI, as it is difficult to ascertain whether children have an underlying primary or secondary bacterial infection and withholding or delaying treatment where indicated poses significant risk.[3] Clinical prediction scores for pneumonia have low specificity in children,[4 5] and infection on standard microbiological tests is confirmed in as little as 22% of treated LRTI.[6 7]

Rapid diagnostic tests have the potential to reduce untargeted antimicrobial use.[8] Most currently available molecular respiratory diagnostic panels include common viruses and a limited number of atypical bacterial pathogens such as *Legionella pneumophila* and *Mycoplasma pneumoniae* which are difficult to grow on culture.[9 10] Some, such as the FilmArray Pneumonia Panel (BioFire Diagnostics, Utah,

USA) and Curetis Unyvero pneumonia cartridge (Holzgerlingen, Germany) offer the advantage of being able to be undertaken as a point of care test with sufficient staffing and training. However, they are constrained to testing manufacturer defined molecular targets.[11–13]

The TaqMan array card (TAC) (Thermo Fisher Scientific, California) is a probe-based quantitative PCR (qPCR) assay undertaken in a 384 well plate format. Advantages of TAC are the ability to customise molecular targets—allowing new targets to be incorporated according to local epidemiology, and the ability to run multiple samples on a single card improving cost efficiency.[14] This may include up to eight clinical samples per card depending on the number of targets sought per sample.

TAC performs reliably on deep respiratory samples obtained from adults with suspected ventilator associated pneumonia.[15] In children, studies of TAC as an LRTI diagnostic are limited. On extensive search of medical literature, we identified three key studies of TAC in children with suspected pneumonia. One used nasopharyngeal/oropharyngeal and sputum in hospitalised children.[16] The second study obtained endotracheal tube (ETT) aspirate samples in 25 children with suspected hospital acquired pneumonia with a limited TAC inclusive of seven bacteria.[17] The third study tested expectorated sputum samples with an eight bacterial pathogen TAC.[18] TAC improved diagnostic yield of samples in all studies,[16–18] but was not used for diagnostic purposes. Obtaining true deep respiratory samples is invasive and technically difficult in children, requiring advanced airway management. These studies used less invasively collected samples as a proxy for bronchoalveolar sampling of the lower respiratory tract. Given the heavy use of systemic antimicrobial therapy in the care of critically ill children, evaluation of TAC to guide antimicrobial therapy requires further evaluation. We are, therefore, undertaking this study to understand the performance and impact of TAC in the PICU setting.

TAC has been evaluated for a number of other paediatric applications, including bloodstream, central nervous system and upper respiratory tract infection,[19–24] but these studies did not evaluate the utility of TAC in clinical diagnoses. These studies demonstrated high sensitivity and specificity of TAC, but it is difficult to extrapolate this into the clinical setting due to variation in molecular targets selected; regional microbiology; and key differences in clinical practice such as timing of antimicrobial administration.

TAC interpretation requires an approach that is distinct from routinely performed investigations such as culture on which only 1–2 predominant species are generally reported. The TAC array may identify several potential pathogens. Identifying multiple micro-organisms in the lungs may be helpful; however, the interpretation can be challenging for clinicians to determine antimicrobial prescriptions. In the case of hospital-acquired LRTI, this may be due to infiltration of the respiratory tract by bacteria from the dysbiotic intestinal microbiome.

Community-acquired LRTI may also have several pathogens, with co-infection by bacterial and viral pathogens a recognised problem in critically ill children.[25] The precise identification of multiple organisms in the lungs in parallel should in theory help to guide the use of antimicrobial therapy, but at a clinical level it demands an understanding of how to interpret the data from molecular pathogen assays. To robustly evaluate TAC, comparison of the assay to a culture-independent technique such as metagenomics can identify microorganisms that may have been eliminated due to prior antimicrobial therapy.

While large-scale studies may assess individual molecular targets included on TAC for their performance, a more holistic approach of assessing an entire multipathogen array, including targets at genus and species level, will give an indication of its clinical application.

## HYPOTHESIS
TAC will provide greater sensitivity and a faster turnaround time than standard microbiology tests for the diagnosis of LRTI in critically unwell children.

## METHODS AND ANALYSIS
### Setting
Patients will be enrolled to the study in a 13 bedded general PICU at Addenbrooke's Hospital, Cambridge, England. The PICU manages neurosurgical and trauma cases but does not support extracorporeal membrane oxygenation or cardiac surgical patients.

### Eligibility criteria
The eligibility criteria are:
1. The child is aged less than 18 years old.
2. The child is receiving mechanical ventilation at the time of enrolment.
3. The child is commencing or already receiving antimicrobial therapy to treat suspected or confirmed LRTI.
   The exclusion criteria are:
1. The patient has a non-survivable illness and is no longer on an active treatment pathway.
2. The child is <37 weeks corrected gestation.

These criteria ensure the patient is able to have samples obtained via non-bronchoscopic bronchoalveolar lavage (NB-BAL). Enrolment of children based on antimicrobial prescription for LRTI has been selected rather than the use of clinical features for pneumonia, given there is no consistent and reliable clinical scoring system for this condition.[26 27] Premature infants have been excluded from enrolment due to the sampling procedure not being tested, to our knowledge, in this group of patients, and these infants having distinct pneumonia aetiology that would require separate evaluation.

### Selection of participants
All children admitted to PICU at the study centre will be screened for enrolment into the study by nursing

$$N = \frac{p_0 q_0 \left\{ z_{1-\alpha/2} + z_{1-\beta} \sqrt{\frac{p_1 q_1}{p_0 q_0}} \right\}^2}{(p_1 - p_0)^2}$$

$$q_0 = 1 - p_0$$
$$q_1 = 1 - p_1$$

$$N = \frac{0.22 * 0.78 \left\{ 1.96 + 0.84 \sqrt{\frac{0.352*0.648}{0.22*0.78}} \right\}^2}{(0.352 - 0.22)^2}$$

$$N = 85$$

$p_0$ = proportion (incidence) of population
$p_1$ = proportion (incidence) of study group
$N$ = sample size for study group
$\alpha$ = probability of type I error (usually 0.05)
$\beta$ = probability of type II error (usually 0.2)
$z$ = critical Z value for a given $\alpha$ or $\beta$

**Figure 1** Power calculation. Calculation generated with ClinCalc.[28]

and medical staff working on the unit. A deferred consent process for up to 48 hours will allow carers to be approached sensitively by the research team, while ensuring samples are obtained within a reasonable time frame to maximise yield. Consent will be obtained via written, electronic and verbal formats via research nursing staff, and permission also obtained for COVID-19-related work (online supplemental appendix A and B). Co-enrolment will be permitted in this study so long as there is no impact on the primary outcomes of either project.

### Sample size measurement

This study aims to achieve a relative increase in the sensitivity of lower respiratory tract culture using TAC by 60%. This is a conservative target, given previous paediatric evaluations of TAC have had a relative increase in detection by >100% in comparison to routine investigations.[16 17] However, this study is distinct in its sampling approach, patient population and TAC configuration, hence outcomes are difficult to estimate based on available literature.

Previous study on this unit has identified that 22% of cases of possible LRTI are confirmed on culture.[7] To increase bacterial confirmation by 60%, a total of 85 patients are required as per power calculation (figure 1).[28] The total has been rounded up to 100 to account for possible sampling related issues that may occur.

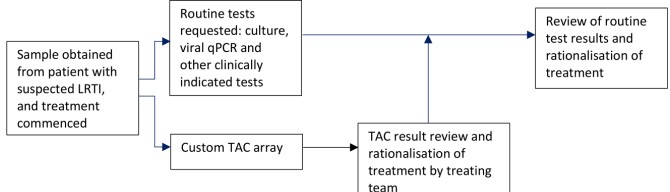

**Figure 2** Workflow. LRTI, lower respiratory tract infection; qPCR, quantitative PCR; TAC, Taqman Array Card.

### Intervention

Diagnostic TAC will be undertaken on NB-BAL sample obtained as soon as possible following the time of enrolment (figure 2). The same sample will be used for standard microbiology culture in all patients; however, the clinical team will determine whether viral qPCR panel, extended fungal or mycobacterial cultures or investigations are also indicated. Samples will only be processed during business hours Monday-Friday due to availability of research and laboratory staff.

Samples will be obtained as part of standard practice by physiotherapy, nursing and medical staff that routinely perform the sampling procedure. The results of the test will be reported alongside routine tests, and impact of these results will be investigated. Positive results will be reported with their corresponding cycle threshold (Ct) values, to assist interpretation where a detection may either be a true pathogen or pathobiont. Standard practice in this PICU is for all patients to be reviewed in twice weekly microbiology rounds. In these meetings, the patient's clinical presentation, progress, biochemical and microbiology results, and treatment is reviewed. The clinical team have a low threshold in seeking input from the microbiology team, which provide a 24 hour on-call service. This multidisciplinary approach will assist the clinical team in situations where there are detections on TAC of unclear significance.

### Study objectives

#### Primary

1. Determine the sensitivity and specificity of TAC in the detection of lower respiratory bacterial pathogens.
2. Compare time to diagnosis of TAC vs standard microbiology culture in the diagnosis of LRTI.

#### Secondary

1. Describe the micro-organisms detected on TAC that were not detected using standard diagnostic tests.
2. Describe the lung microbiome of critically ill children with LRTI using 16S rRNA gene sequencing.
3. Understand the impact of TAC on total antimicrobial prescriptions in PICU by total prescriptions and by antimicrobial class.
4. Describe the impact of TAC on antimicrobial decision making according to PICU consultants.
5. Describe the host inflammatory response to LRTI in the lung and blood according to pathogen present.

6. Describe the experience of a tertiary PICU in the implementation of TAC in the clinical setting.

## Outcome measures
### Primary
1. TAC result (Ct value).
2. Respiratory culture result (colony forming units/mL).
3. Respiratory viral panel result (Ct value).
4. Time to result (hours).

### Secondary
1. TAC result (Ct value).
2. Blood culture result.
3. Total days of antimicrobial prescriptions in PICU—% total days of PICU admission and days free of antimicrobial therapy in PICU at 28 days.
4. Reported n (%) of change in antimicrobial prescription type and duration, and n (%) of children able to move out of side rooms due to exclusion of infectious pathogens.
5. Days free of mechanical ventilation at 28 days following PICU admission.
6. Days free of PICU at 28 days following PICU admission.
7. 16S rRNA gene sequencing result—n (%) of total reads, Shannon's diversity index
8. Cytokine concentration (pg/mL) in NB-BAL samples and blood.
9. Semistructured interview thematic analysis—description of implementation and interpretation considerations of TAC in the PICU.

## SAMPLING AND LABORATORY PROCEDURES
### NB-BAL sampling
Deep respiratory samples will be obtained at the time of enrolment via NB-BAL, with some modifications made to the original procedure given SARS-CoV-2 (online supplemental appendix C and D). Saline lavage volume instilled will be 1 mL/kg of patient weight to a maximum of 10 mL. Saline is delivered via the in-line suction catheter system and the sample collected in a universal container via a sputum trap. This approach minimises risk of aerosolisation of pathogens such as SARS-CoV-2. Samples will be immediately delivered to the microbiology containment level 3 laboratory and stored at 4°C until staff are available to process the samples, between 8:00 and 17:00 hours weekdays. Samples will be split into aliquots for microbiology culture, nucleic acid extraction and any additional clinical tests required.

### Nucleic acid extraction
Up to 750 µL of sample will be added to a 2 mL microtube containing a mixture of ceramic beads with 750 µL of L6 buffer (Qiagen). A minimum of 100 uL will be required for extraction and brought up to 750 µL with nuclease free water if low volume. The sample will then be vortexed and incubated for 10 min. Samples will then be processed using an EZ1 virus mini kit (V.2.0) using an EZ1 advanced

XL (Qiagen) in up to 14 samples per run including an L6 buffer-RNase free water negative control.[29]

### TaqMan array card
A custom screening panel was developed and validated via an adult intensive care study that took place in this institution.[30] Retrospective review of organisms identified on routine tests for severe LRTI in this PICU demonstrated good coverage by this panel.[7] The targets of the panel are as shown in table 1. Many pathogens are assigned two targets to reduce false positive test interpretations. A target for the MecA gene has been included as it is commonly present in methicillin-resistant *Staphylococcus aureus*. The TAC includes endogenous control RNase P, internal control MS2 and 18S rRNA gene. TAC configuration was altered on 5 February 2021, to include SARS-CoV-2 targets, and opportunistically incorporate *Leptospira* as this organism was of relevance to the adult ICU service also using the cards. Using the same card for the adult and paediatric service allows samples to be processed in batches of up to four, reducing waste of empty lanes in the card, and minimises laboratory handling time.

For each sample, 50 µL of total nucleic acid is added to 50 µL of TaqMan Fast Virus 1-step mastermix (Thermo Fisher) and 100 µL of RNase free water. A total of 98 µL is then added to two lanes of the array. Each lane comprises 48 molecular targets, with the array configured to 96 targets of interest. Therefore, four patient samples are loaded into two lanes each for a total of eight lanes per plate.

The RT-PCR will be undertaken on a QuantStudio 7 Flex (Thermo Fisher) according to the following validated protocol: 50°C for 5 min, 95°C for 20 s, 45 cycles of 95°C for 1 s, 60°C for 20 s.[15] qPCR Ct values with clear amplification curves will be reported and documented on the electronic medical record.

### Conventional pathogen testing
Samples will be processed according to Public Health England laboratory standards. For patients that are not immunocompromised, standard media will be used. The sample will be inoculated on chocolate agar and incubated at 35°C–37°C supplemented with 5%–10% $CO_2$.[31] If the patient is immunocompromised, supplementary media will be used, and MacConkey agar and Mannitol salt/Chromogenic agar will be used with the sample incubated in air.[31] Significant growth constitutes >$10^4$ cfu/mL on NB-BAL samples, or >$10^5$ cfu/mL on ETT aspirate. Bacterial organisms will be identified to species or genus level using Matrix-Assisted Laser Desorption/Ionisation Time of Flight (MALDI-TOF) mass spectrometry (Bruker Daltonics, Coventry, UK). Antimicrobial susceptibility testing will be performed using disc diffusion following European Committee on Antimicrobial Susceptibility Testing (EUCAST) guidelines.[32]

An in-house multiplex reverse transcription (RT)-PCR assay will be used for testing NPA and NB-BAL samples for common respiratory viruses: adenovirus, enterovirus,

**Table 1** Molecular targets of the TaqMan diagnostic card for lower respiratory tract infection

| Type | Target | No of targets |
|---|---|---|
| Bacterial | 16S rRNA gene* | 2 |
| | *Acinetobacter baumannii* | 3 |
| | *Bacteroides fragilis* | 1 |
| | *Bordetella pertussis* | 2 |
| | *Chlamydia pneumoniae* | 1 |
| | *Chlamydia psittaci* | 1 |
| | *Coxiella burnetti* | 1 |
| | *Elizabethkingia meningoseptica* | 2 |
| | *Enterobacter cloacae* | 2 |
| | *Enterobacteriaceae* | 1 |
| | *Enterobacteriaceae Proteus* | 1 |
| | *Enterococcus faecalis** | 1 |
| | *Enterococcus faecium** | 2 |
| | *Escherichia coli* | 2 |
| | *Haemophilus influenzae* | 2 |
| | *Klebsiella pneumoniae* | 2 |
| | *Legionella pneumophilia* | 1 |
| | *Legionella* spp | 1 |
| | *Moraxella catarrhalis* | 1 |
| | *Morganella morganii* | 1 |
| | *Mycobacterium* spp | 1 |
| | *Mycoplasma pneumoniae* | 2 |
| | *Mycobacterium tuberculosis* | 2 |
| | *Neisseria meningitidis* | 1 |
| | *Pseudomonas aeruginosa* | 2 |
| | *Serratia marcescens* | 2 |
| | *Staphylococcus aureus* | 2 |
| | *Staphylococcus*-coagulase negative | 1 |
| | *Staphylococcus epidermidis* | 1 |
| | *Staphylococcus*-PVL toxin | 1 |
| | *Stenotrophomonas maltophilia* | 1 |
| | *Streptococcus pneumoniae* | 2 |
| | *Streptococcus pyogenes* | 2 |
| | *Streptococcus* spp | 2 |
| Viral | Adenovirus | 2 |
| | Bocavirus | 1 |
| | Cytomegalovirus | 1 |
| | Epstein Barr virus | 1 |
| | Enterovirus | 2 |
| | Herpes simplex virus | 2 |
| | Human coronavirus NL63 | 1 |
| | Human coronavirus OC43/HKU1 | 1 |
| | Human coronavirus OC43 | 1 |

Continued

**Table 1** Continued

| Type | Target | No of targets |
|---|---|---|
| | Human coronavirus 229E | 1 |
| | Human metapneumovirus | 1 |
| | Human parainfluenza virus 1 | 2 |
| | Human parainfluenza virus 2 | 1 |
| | Human parainfluenza virus 3 | 2 |
| | Human parainfluenza virus 4 | 1 |
| | Influenza A | 2 |
| | Influenza A-H1 (2009) | 1 |
| | Influenza A-H3 | 1 |
| | Influenza B | 2 |
| | Parechovirus | 1 |
| | Respiratory syncytial virus (any) | 1 |
| | Respiratory syncytial virus A | 1 |
| | Respiratory syncytial virus B | 1 |
| | Rhinovirus | 2 |
| Fungal | *Aspergillus* 28S rRNA gene | 2 |
| | *Aspergillus fumigatus* | 1 |
| | *Candida albicans* | 1 |
| | *Candida* species | 1 |
| | Fungal 18S rRNA gene | 1 |
| | Pneumocystis jirovecii | 1 |
| AMR gene | *mecA* | 1 |
| Controls | 18S rRNA gene | 1 |
| | MS2 internal control | 2 |
| | RNase P internal control | 1 |
| **Total wells** | | **96** |

*In enrolments occurring from 5 February 2021 onwards, these targets were replaced to include three targets for SARS-CoV-2, 1 target for *Leptospira* and an additional target for legionella species.

human metapneumovirus, influenza A and B viruses, parainfluenza virus, rhinovirus and respiratory syncytial virus. This assay is only requested when determined to be of relevance by the clinical team.

If clinically suspected, presence of *Aspergillus* spp will be tested by culture on Sabouraud dextrose agar with chloramphenicol. Serum will also be tested for the presence of galactomannan antigen. If clinically suspected PCR assay will be used to detect presence of *Pneumocystis jirovecii*.

### 16S rRNA gene sequencing

Nucleic acid samples will be quantified using a Qubit 4 fluorometer and Qubit dsDNA HS assay kit (Q32854) (ThermoFisher Scientific, Waltham, Massachusetts, USA). DNA fragment size and quality will be assessed using an Agilent TapeStation 4200 (Agilent, Santa Clara, California, USA).

After quantification, a sequencing library will be prepared using a 16S rRNA gene Barcoding Kit (SQK—16S024), then loaded onto a FLO-MIN106D R9.4.1 flow cell. Sequencing will be undertaken using a MinION nanopore sequencing device (Oxford Nanopore Technologies, UK). Base calling will be undertaken using the EPI2ME platform and data will be stored as FASTQ files. Demultiplexing will be completed with guppy_barcoder V.5.0.11, adapters will be trimmed with porechop 0.2.4, and reads filtered with NanoFilt V.2.8.0. Filtered reads will be classified using Kraken2.[33] Results will be for research purposes only and not distributed to the clinical team.

### Blood sampling
Blood will be obtained for research if the child is undergoing venepuncture for clinical purposes or if there is an existing vascular access device. The maximum volume obtained will be 1 mL/kg to a maximum 10 mL in keeping with WHO guidelines.[34] Collected blood will be stored in a 4°C refrigerator on the ward until the research team are able to process samples. EDTA tubes will be spun at 1300 g for 10 min at 4°C, and aliquots stored at −20°C until batch processing.

### Cytokine assay
Cytokines will be quantified from NB-BAL and plasma samples using a Bio-Plex Pro Human Cytokine Screening 48-plex kit (Bio-Rad) using standard methods.[35] This assay will be undertaken on aliquots of NB-BAL stored at −80°C until batch processing. NB-BAL will be undiluted while plasma will be diluted with standard diluent HB as per manufacturer recommendations dependent on limits of detection according to the generated standard curves.

### Survey and semistructured interviews
Senior medical staff on the PICU will be surveyed regarding their antimicrobial decision making after TAC result becomes available (online supplemental appendix E). Survey invitation will be sent via Research Eletronic Data Capture (REDCap) to the hospital email address of the duty PICU consultant when the TAC result becomes available.

Semistructured interviews will be undertaken with PICU nursing and medical staff at the end of the study. They will be advertised using posters and via the internal newsletter to staff. Voluntary consent will be obtained (online supplemental appendix F and G). Sessions will be facilitated by a member of the research team and recorded. Themes and descriptions will be analysed using NVivo V.11.[36] These interviews will capture the experience of staff in implementing TAC into clinical practice, the benefits and downsides to the test, and interpretation of the test.

### Patient and public involvement
Carers of children admitted to PICU identified rapid molecular diagnostic tests for early rationalisation of antimicrobial therapy to be a high priority area for research. They ranked this fourth of 73 potential national priority areas for PICU research in a 2019 multicentre survey.[37] As this study is investigating acute clinical decision making of critical care clinicians, patients were not involved in the design of scientific study methodology. The research team will be engaging interested families who have had children admitted to this PICU for their input in the dissemination of research findings and for additional feedback on the design of patient information and consent materials.

### Study status
Enrolment commenced on 10 April 2020 and enrolment is expected to be completed prior to March 2023.

## DATA MANAGEMENT
### Database
Data will be obtained from the electronic medical record and recorded on REDCap, a secure data management system which will be hosted by the University of Cambridge.[38] Data collected will include demographics, physiological parameters, paediatric index of mortality 3 score,[39] treatment received and investigation results from the time of presentation to hospital for the acute illness to discharge.

### Statistical analysis plan
Data will be analysed in R.[40] Demographic data will be reported using simple descriptive statistics including mean and standard deviation (SD). For comparisons of demographic data between groups, skewed data using non-parametric tests including Mann-Whitney U test and Kruskal-Wallis test. Normally distributed data will be compared with Student's t-tests. Group correlations of quantitative data will be tested with Spearman correlation test.

All TAC detections from NB-BAL samples will be reported using the mean and range of Ct values for each molecular target. This will be compared with growth on culture in cfu/mL or Ct value for routine qPCR testing. A positive TAC result will be considered a Ct of 32 or below, which was found to correlate with microbiology culture thresholds in a previous investigation.[30]

Time to reportable results will be recorded as the time taken from collection time of the sample, which will be entered by nursing staff into the electronic medical record. The comparison of time to result of TAC versus culture will be assessed with Wilcoxon signed ranks test.

Cytokine results will be analysed using a support-vector machine approach to explore profiles of different infections.

## ETHICS AND DISSEMINATION
### Ethics and registration
The study is jointly sponsored by Cambridge University Hospitals NHS Foundation Trust and the University of Cambridge. The study was approved by the Yorkshire and

the Humber-Bradford Leeds Research Ethics Committee on 26 March 2020 (REC reference 20/YH/0089). The amended, current version of the protocol (8.0) was approved by the REC on 2 July 2021. Amendments to this study have primarily been a result of the SARS-CoV-2 pandemic. Additional work packages investigating SARS-CoV-2 infection in children under the overarching RASCALS project are subject to separate protocol and analysis that are not outlined in this paper.

Protocol amendments will be communicated to participants when changes are made that result in a change in procedures for the individual. Participants will be pseudonymised via creation of a sequential study ID.

## INFORMED CONSENT PROCEDURES

Consent will be obtained in written, electronic or verbal format as approved by the ethics committee. While written consent is preferred, alternative consent procedures were introduced due to the COVID-19 pandemic. Deferred consent for up to 48 hours will be permitted to allow time critical samples to be obtained, while also ensuring families are approached sensitively in the PICU.

## DISSEMINATION

Findings from this project will be reported in peer-reviewed journals and international conferences.

**Acknowledgements** The authors would like to thank the staff at Addenbrooke's Hospital for supporting the development and implementation of this study; Helen Starace and Colin Hamilton (physiotherapists) for assisting with developing NB-BAL sampling procedure; Adam Palmer (PICU data manager) for providing admissions data; and Claire Jenkins (scientist) for assisting with sample workflow.

**Contributors** JAC, MDC, ACM, VN, MET, SB and NP conceived the study design; JAC, MDC, ACM, VN, MET, SB and NP designed the study protocol; JAC, IRLK, MDC, FK, JBS, MM, and SB developed laboratory methods for the study. JAC, DW, ED, SA and DI developed sampling procedure protocol; JAC, IRLK, FK, ACM, RB, TG, ZZ, SB and NP developed the statistical analysis plan; JAC, IRLK, MDC, FK, DW, ED, ACM, VN, RB, TG, SA, ZZ, SB and NP drafted this manuscript.

**Funding** This project is funded by the Addenbrooke's Charitable Trust, Cambridge University Hospitals (900240) (JAC, NP, MET and SB) which will provide funding for consumables, 16S rRNA gene sequencing of samples and inflammatory profiling; in addition to the NIHR Cambridge Biomedical Research Centre. The authors also receive support from the Gates Cambridge Trust (OPP1144) (JAC); the Academy of Medical Sciences (MET); Wellcome Trust [215515] (SB); Wellcome Trust Clinical Research Career Development Fellowship (WT 2055214/Z/16/Z) (ACM) MRC Clinician Scientist Fellowship [MR/V006118/1] (ACM); and Action Medical Research (NP, SB, MET) (GN2751, GN2903).This work was supported, in whole or in part, by the Bill & Melinda Gates Foundation (OPP1144). Under the grant conditions of the Foundation, a Creative Commons Attribution 4.0 Generic License has already been assigned to the Author Accepted Manuscript version that might arise from this submission.

**Competing interests** MDC is the inventor on a patent held by the Secretary of State for Health (UK government) EP2788503, which covers some of the genetic sequences used in this study. VN is a founder, director and shareholder in Cambridge Infection Diagnostics (CID) which is a commercial company aimed at developing molecular diagnostics in infection and antimicrobial and AMR stewardship. ACM, SB and ED are members of the Scientific Advisory Board of Cambridge Infection Diagnostics (CID). TG has received a research grant from Shionogi. All other authors declare no conflict of interest.

**Patient consent for publication** Not applicable.

**Provenance and peer review** Not commissioned; externally peer reviewed.

**ORCID iDs**
John Alexander Clark http://orcid.org/0000-0001-6916-9195
M Estée Török http://orcid.org/0000-0001-9098-8590

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
