## [Reviewer comments · BMJ Open]

ARTICLE DETAILS

TITLE (PROVISIONAL)	Rapid Assay for Sick Children with Acute Lung infection Study (RASCALS) – diagnostic cohort study protocol
AUTHORS	Clark, John; Kean, Iain; Curran, Martin; Khokhar, Fahad; White, Deborah; Daubney, Esther; Conway-Morris, Andrew; Navapurkar, Vilas; Bartholdson-Scott, Josefin; Maes, Mailis; Bousfield, Rachel; Gouliouris, Theodore; Agrawal, Shruti; Inwald, David; Zhang, Zhenguang; Torok, Mili; Baker, Stephen; Pathan, Nazima

VERSION 1 – REVIEW

REVIEWER	Caroselli, Costantino INRCA-IRCCS
REVIEW RETURNED	12-Sep-2021

GENERAL COMMENTS	Dear Author, I read your paper with great interest and attention. I find this work interesting but I would suggest to add to your paper a discussion and conclusions. I also suggest that you avoid attaching all consent forms used, specifying that informed consent has been correctly obtained from all study participants.
---

REVIEWER	Berkley, James KEMRI/Wellcome Trust Research Programme, Clinical Research
REVIEW RETURNED	05-Oct-2021

GENERAL COMMENTS	The protocol addresses an important topic with the potential to improve timely care. The protocol is clearly written, and presents a strong rationale for the initial TAC assay and a straightforward study design. The inclusion of a clinical team survey is a helpful implementation science component. A few paediatric studies using TAC are referenced – there are quite a few others, including several done in low- and middle-income settings in pneumonia and in neonates. An issue that is raised in this literature is that molecular methods can lack specificity. I think this would be important to comment on in the background, especially in relation to results may ultimately be used to guide antimicrobial therapy. The sample size is based on a large increase in bacterial pathogen detection by 60% (it is not clear in the text if this is an absolute or relative increase) and it would be helpful to provide references to support this being a realistic target.
--

	As mentioned above, giving an indication of the study design's ability to report specificity is also important as if increasing detection comes with a greater proportion of false positives (even with paired targets) then that may influence treatment decisions. Information is given to carers about later samples at home at 4 weeks, however this does not appear in the abstract, hypothesis, objectives, outcome measures, 'intervention' or analysis plan - I suggest updating these to provide the rationale, specific objectives and proposed analysis.
--	---

VERSION 1 – AUTHOR RESPONSE

Reviewer: 1

Prof. Costantino Caroselli, INRCA-IRCCS

Comments to the Author:

Dear Author,

I read your paper with great interest and attention. I find this work interesting but I would suggest to add to your paper a discussion and conclusions [NOTE FROM THE EDITORS: please feel free to rebut this request - Discussion sections are optional for protocol papers, and Conclusions sections are not appropriate for protocols].

On completion of this study, the research team will publish findings in a peer-reviewed journal. In this publication we will provide a broader discussion of the impacts, limitations and context of this research and provide conclusions of what this research contributes to the scientific literature. Context of the present protocol paper is presented in the 'introduction' section and additional discussion has been added following peer review.

I also suggest that you avoid attaching all consent forms used, specifying that informed consent has been correctly obtained from all study participants. [NOTE FROM THE EDITORS: please feel free to rebut the request to delete the supplemental files containing the model consent forms, these are fine to include].

Consent forms have been included to provide transparency in the information delivered to participants.

A few paediatric studies using TAC are referenced – there are quite a few others, including several done in low- and middle-income settings in pneumonia and in neonates. An issue that is raised in this literature is that molecular methods can lack specificity. I think this would be important to comment on in the background, especially in relation to results may ultimately be used to guide antimicrobial therapy.

Additional papers have been referenced regarding use of TAC in the paediatric setting. Limitations of existing studies are – firstly the sampling technique, which assumes that upper respiratory swabs, expectorated and induced sputum and ETT aspirates can reliably represent the lower respiratory tract. Secondly, these studies have largely been conducted in the laboratory setting and have not influenced clinical decision making. Thirdly, patients in these studies have frequently been hospitalised but not required intensive care, where antimicrobial use is most heavily used and where illness severity is greatest. Use of culture in intensive care patients is problematic, as they have frequently received antimicrobial treatment before the culture is performed, hence has poor sensitivity.

Specificity has been added as a primary study objective – however, will require interpretation with caution. This study will be evaluating performance of this custom TAC holistically, rather than the performance of each individual target. Given some targets on the array will be low prevalence causes of LRTI, it is unlikely that we will have sufficient representation of true infection by these micro-organisms to adequately evaluate them individually – this would require a multi-centre study design for sufficient power. This project will provide data to support the design of a larger evaluation such as this.

In the situation that there are detections on TAC with high cycle threshold values or where detection/s on TAC are of unclear significance (potentially due to specificity of the test), a multi-disciplinary approach will be taken between the clinical and microbiology teams to determine action required. Details regarding the teams' approach have been added to the manuscript.

The sample size is based on a large increase in bacterial pathogen detection by 60% (it is not clear in the text if this is an absolute or relative increase) and it would be helpful to provide references to support this being a realistic target.

Additional justification for relative increase in detection provided in section 'Sample size measurement'

As mentioned above, giving an indication of the study design's ability to report specificity is also important as if increasing detection comes with a greater proportion of false positives (even with paired targets) then that may influence treatment decisions.

Additional detail has also been provided under 'intervention' relating to how study results will be provided to clinicians and dealing with possible false positives.

Information is given to carers about later samples at home at 4 weeks, however this does not appear in the abstract, hypothesis, objectives, outcome measures, 'intervention' or analysis plan - I suggest updating these to provide the rationale, specific objectives and proposed analysis.

The RASCAL study was under ethics review at an early phase of the COVID-19 pandemic. Given the study was designed to collect samples from critically ill children with respiratory illness, it was able to be amended to also investigate children that presented with COVID-19 related illness. Study amendments were therefore approved to investigate this separate cohort of children, under the same overarching protocol. The COVID-19 related illness enrolments to RASCALS will be subjected to alternative analysis and reporting to the 100 LRTI enrolments in the present protocol paper. This will ensure findings are reported clearly, and with attention to the primary aims of the study. The follow-up samples in the 4 weeks following enrolment belong only to those children enrolled to this COVID-19 cohort, hence are not discussed here.

VERSION 2 – REVIEW

REVIEWER	Berkley, James KEMRI/Wellcome Trust Research Programme, Clinical Research
REVIEW RETURNED	11-Nov-2021
GENERAL COMMENTS	My comments have been addressed